# DEEP ASYMMETRIC MULTI-TASK FEATURE LEARNING

## ABSTRACT

We propose Deep Asymmetric Multitask Feature Learning (Deep-AMTFL) which can learn deep representations shared across multiple tasks while effectively preventing negative transfer that may happen in the feature sharing process. Specifically, we introduce an asymmetric autoencoder term that allows reliable predictors for the easy tasks to have high contribution to the feature learning while suppressing the influences of unreliable predictors for more difficult tasks. This allows the learning of less noisy representations, and enables unreliable predictors to exploit knowledge from the reliable predictors via the shared latent features. Such asymmetric knowledge transfer through shared features is also more scalable and efficient than inter-task asymmetric transfer. We validate our Deep-AMTFL model on multiple benchmark datasets for multitask learning and image classification, on which it significantly outperforms existing symmetric and asymmetric multitask learning models, by effectively preventing negative transfer in deep feature learning.

## 1 INTRODUCTION

Multi-task learning (Caruana, 1997) aims to improve the generalization performance of the multiple task predictors by jointly training them, while allowing some kinds of knowledge transfer between them. One of the crucial challenges in multi-task learning is tackling the problem of *negative transfer*, which describes the situation where accurate predictors for easier tasks are negatively affected by inaccurate predictors for more difficult tasks. A recently introduced method, Asymmetric Multi-task Learning (AMTL) (Lee et al., 2016), proposes to solve this negative transfer problem by allowing asymmetric knowledge transfer between tasks through inter-task parameter regularization. Specifically, AMTL enforces the task parameters for each task to be also represented as a sparse combination of the parameters for other tasks, which results in learning a directed graph that decides the amount of knowledge transfer between tasks.

However, such inter-task transfer model based on parameter regularization is limited in several aspects. First of all, in most cases, the tasks exhibit relatedness to certain degree, but the model parameter for a task might not be reconstructed as a combination of the parameter for other tasks, because the tasks are only partly related. Consider the example in Fig. 1(b), where the task is to predict whether the given image has any of the three animal classes. Here, the three animal classes are obviously related as they share a common visual attribute, *stripe*. Yet, we will not be able to reconstruct the model parameter for class hyena by combining the model parameters for class tiger and zebra, as there are other important attributes that define the hyena class, and the *stripe* is merely a single attribute among them that is also shared by other classes. Thus, it is more natural to presume that the related tasks leverage a common set of latent representations, rather than considering that a task parameter is generated from the parameters for a set of relevant tasks, as assume in inter-task transfer models.

Moreover, AMTL does not scale well with the increasing number of tasks since the inter-task knowledge transfer graph grows quadratically, and thus will become both inefficient and prone to overfitting when there are large number of tasks, such as in large-scale classification. While sparsity can help reduce the number of parameters, it does not reduce the intrinsic complexity of the problem.

Finally, the inter-task transfer models only store the knowledge in the means of learned model parameters and their relationship graph. However, at times, it might be beneficial to store what has

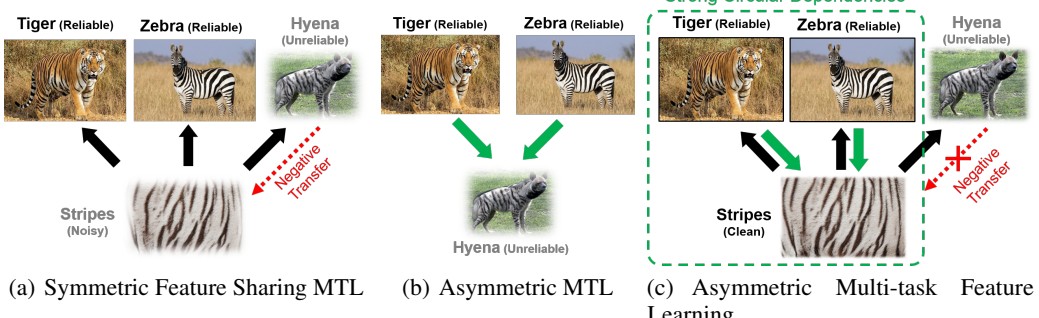

(a) Symmetric Feature Sharing MTL  (b) Asymmetric MTL  (c) Asymmetric Multi-task Feature Learning

Figure 1: **Concept.** (a) Feature-sharing multi-task learning models such as Go-MTL suffers from negative transfer from unreliable predictors, which can result in learning noisy representations. (b) AMTL, an inter-task transfer asymmetric multi-task learning model that enforces the parameter of each task predictor to be generated as a linear combination of the parameters of relevant task predictors, may not make sense when the tasks are only partially related. (c) Our asymmetric multi-task feature learning enforces the learning of shared representations to be affected only by reliable predictors, and thus the learned features can transfer knowledge to unreliable predictors.

been learned, in the form of explicit representations which can be used later for other tasks, such as transfer learning.

Thus, we resort to the multi-task feature learning approach that aims to learn latent features, which is one of the most popular ways of sharing knowledge between tasks in the multi-task learning framework (Argyriou et al., 2008; Kumar & Daume III, 2012), and aim to prevent negative transfer under this scenario by enforcing asymmetric knowledge transfer. Specifically, we allow reliable task predictors to affect the learning of the shared features more, while downweighting the influences of unreliable task predictors, such that they have less or no contributions to the feature learning. Figure 1 illustrates the concept of our model, which we refer to as asymmetric multi-task feature learning (AMTFL).

Another important advantage of our AMTFL, is that it naturally extends to the feature learning in deep neural networks, in which case the top layer of the network contains additional weight matrix for feed-back connection, along with the original feed-forward connections, that allows asymmetric transfer from each task predictor to the bottom layer. This allows our model to leverage state-of-the-art deep neural network models to benefit from recent advancement in deep learning.

We extensively validate our method on a synthetic dataset as well as seven benchmark datasets for multi-task learning and image classification using both the shallow and the deep neural network models, on which our models obtain superior performances over existing symmetric feature-sharing multi-task learning model as well as the inter-task parameter regularization based asymmetric multi-task learning model.

Our contributions are threefold:

- We propose a novel multi-task learning model that prevents negative transfer by allowing asymmetric transfer between tasks, through latent shared features, which is more natural when tasks are correlated but not cause and effect relationships, and is more scalable than existing inter-task knowledge transfer model.

- We extend our asymmetric multi-task learning model to deep learning setting, where our model obtains even larger performance improvements over the base network and linear multi-task models.

- We leverage learned deep features for knowledge transfer in a transfer learning scenarios, which demonstrates that our model learns more useful features than the base deep networks.

## 2 RELATED WORK

**Multitask learning** Multi-task Learning (Caruana, 1997) is a learning framework that jointly trains a set of task predictors while sharing knowledge among them, by exploiting relatedness between participating tasks. Such task relatedness is the main idea on which multi-task learning is based, and there are several assumptions on how the tasks are related. Probably the most common assumption is that the task parameters lie in a low-dimensional subspace. One example of a model based on this assumption is multi-task feature learning (MTFL) (Argyriou et al., 2008), where a set of related tasks learns common features (or representations) shared across multiple tasks. Specifically, they propose to discard features that are not used by most tasks by imposing the $(2, 1)$-norm regularization on the coefficient matrix, and solved the regularized objective with an equivalent convex optimization problem. The assumption in MTFL is rather strict, in that the $(2, 1)$-norm requires the features to be shared across all tasks, regardless of whether the tasks are related or not. To overcome this shortcoming, (Kang et al., 2011) suggest a method that also learns to group tasks based on their relatedness, and enforces sharing only within each group. However, since such strict grouping might not exist between real-world tasks, (Kumar & Daume III, 2012) and (Maurer et al., 2012) suggest to learn overlapping groups by learning latent parameter bases that are shared across multiple tasks.

**Asymmetric Multitask learning** The main limitation in the multi-task learning models based on common bases assumption is that they cannot prevent negative transfer as shared bases are trained without consideration of the quality of the predictors. To tackle the problem, asymmetric multi-task learning (AMTL) (Lee et al., 2016) suggests to break the symmetry in the knowledge transfer direction between tasks. It assumes that each task parameter can be represented as a sparse linear combination of other task parameters, in which the knowledge flows from task predictors with low loss to predictors with high loss. Then, hard tasks could exploit more reliable information from easy tasks, whereas easy tasks do not have to rely on hard tasks when it has enough amount of information to be accurately predicted. The prior assumption of AMTL is thus reasonable. If there is little information to share between tasks, then each task is trained close to single task learning. However, in AMTL, knowledge is transferred from one task to another, rather than from tasks to some common feature spaces. Thus the model is not scalable and also it is not straightforward to apply this model to deep learning. On the contrary, our model is scalable and straightforward to implement into a deep neural network as it learns to transfer from tasks to shared features.

**Autoencoders** Our asymmetric multi-task learning formulation has a sparse nonlinear autoencoder term for feature learning. The specific role of the term is to reconstruct latent features from the model parameters using sparse nonlinear feedback connections, which results in the denoising of the latent features. Autoencoders were first introduced in (Rumelhart et al., 1986) for unsupervised learning, where the model is given the input features as the output and learns to transform the input features into a latent space and then decode them back to the original features. While there exist various autoencoder models, our reconstruction term closely resembles the sparse autoencoder (Ranzato et al., 2007), where the transformation onto and from the latent feature space is sparse. (Hinton & Salakhutdinov, 2006) introduce a deep autoencoder architecture in the form of restricted Boltzmann machine, along with an efficient learning algorithm that trains each layer in a bottom-up fashion. (Vincent et al., 2008) propose a denoising autoencoder, which is trained with the corrupted data as the input and the clean data as output, to make the internal representations to be robust from corruption. Our regularizer in some sense can be also considered as a denoising autoencoder, since its goal is to refine the features through the autoencoder form such that the reconstructed latent features reflect the loss of more reliable predictors, thus obtaining *cleaner* representations.

## 3 ASYMMETRIC MULTI-TASK FEATURE LEARNING

In our multi-task learning setting, we have $T$ different tasks with varying degree of difficulties. For each task $t \in \{1, \ldots, T\}$, we have an associated training dataset $\mathcal{D}_t = \{(\boldsymbol{X}_t, \boldsymbol{y}_t) | \boldsymbol{X}_t \in \mathbb{R}^{N_t \times d}, \boldsymbol{y}_t \in \mathbb{R}^{N_t \times 1}\}$ where $\boldsymbol{X}_t$ and $\boldsymbol{y}_t$ respectively represent the $d$-dimensional feature vector and the corresponding labels for $N_t$ data instances. of multi-task learning is then to jointly train models

for all $T$ tasks simultaneously via the following generic learning objective:

$$\min_{\boldsymbol{W}}. \sum_{t=1}^{T} \mathcal{L}(\boldsymbol{w}_t; \boldsymbol{X}_t, \boldsymbol{y}_t) + \Omega(\boldsymbol{W}). \tag{1}$$

where $\mathcal{L}$ is the loss function applied across the tasks, $\boldsymbol{w}_t \in \mathbb{R}^d$ is the model parameter for task $t$ and $\boldsymbol{W} \in \mathbb{R}^{d \times T}$ is the column-wise concatenated matrix of $\boldsymbol{w}$ defined as $\boldsymbol{W} = [\boldsymbol{w}_1 \, \boldsymbol{w}_2 \cdots \boldsymbol{w}_T]$. Here, the penalty $\Omega$ enforces certain prior assumption on sharing properties across tasks in terms of $\boldsymbol{W}$.

One of the popular assumptions is that there exists a common set of latent bases across tasks (Argyriou et al., 2008; Kumar & Daume III, 2012), in which case the matrix $\boldsymbol{W}$ can be decomposed as $\boldsymbol{W} = \boldsymbol{LS}$. Here, $\boldsymbol{L} \in \mathbb{R}^{d \times k}$ is the collection of $k$ latent bases while $\boldsymbol{S} \in \mathbb{R}^{k \times T}$ is the coefficient matrix for linearly combining those bases. Then, with a regularization term depending on $\boldsymbol{L}$ and $\boldsymbol{S}$, we build the following multi-task learning formulation:

$$\min_{\boldsymbol{L}, \boldsymbol{S}}. \sum_{t=1}^{T} \mathcal{L}(\boldsymbol{L}\boldsymbol{s}_t; \boldsymbol{X}_t, \boldsymbol{y}_t) + \Omega(\boldsymbol{L}, \boldsymbol{S}). \tag{2}$$

where $\boldsymbol{s}_t$ is $t^{th}$ column of $\boldsymbol{S}$ to represent $\boldsymbol{w}_t$ as the linear combination of shared latent bases $\boldsymbol{L}$, that is, $\boldsymbol{w}_t = \boldsymbol{L}\boldsymbol{s}_t$. As a special case of Eq.(2), Go-MTL(Kumar & Daume III, 2012), for example, encourages $\boldsymbol{L}$ to be element-wisely $\ell_2$ regularized and each $\boldsymbol{s}_t$ sparse:

$$\min_{\boldsymbol{L}, \boldsymbol{S}}. \sum_{t=1}^{T} \left\{ \mathcal{L}(\boldsymbol{L}\boldsymbol{s}_t; \boldsymbol{X}_t, \boldsymbol{y}_t) + \mu \left\| \boldsymbol{s}_t \right\|_1 \right\} + \lambda \left\| \boldsymbol{L} \right\|_F^2. \tag{3}$$

On the other hand, it is possible to exploit a task relatedness without the explicit assumption on shared latent bases. AMTL (Lee et al., 2016) is such an instance of multi-task learning, based on the assumption that each task parameter $\boldsymbol{w}_t \in \mathbb{R}^d$ is reconstructed as a sparse combination of other task parameters $\{\boldsymbol{w}_s\}_{s \neq t}$. In other words, it encourages that $\boldsymbol{w}_t \approx \sum_{s \neq t} B_{st} \boldsymbol{w}_s$ where the weight $B_{st}$ of $\boldsymbol{w}_s$ in reconstructing $\boldsymbol{w}_t$, can be interpreted as the amount of knowledge transfer from task $s$ to $t$. Since there is no symmetry constraint on the matrix $\boldsymbol{B}$, AMTL learns asymmetric knowledge transfer from easier tasks to harder ones. Towards this goal, AMTL solves the multi-task learning problem via the following optimization problem:

$$\min_{\boldsymbol{W}, \boldsymbol{B}}. \sum_{t=1}^{T} (1 + \alpha \left\| \boldsymbol{b}_t^o \right\|_1) \mathcal{L}(\boldsymbol{w}_t; \boldsymbol{X}_t, \boldsymbol{y}_t) + \gamma \left\| \boldsymbol{W} - \boldsymbol{W}\boldsymbol{B} \right\|_F^2. \tag{4}$$

where $B_{tt} = 0$ for $t = 1, ..., T$ and $\boldsymbol{B}$'s row vector $\boldsymbol{b}_t^o \in \mathbb{R}^{1 \times T}$ controls the amount of outgoing transfer from task $t$ to other tasks $s \neq t$. The sparsity parameter $\alpha$ is multiplied by the amount of training loss $\mathcal{L}(\boldsymbol{w}_t; \boldsymbol{X}_t, \boldsymbol{y}_t)$, making the outgoing transfer from hard tasks more sparse than those of easy tasks. The second Frobenius norm based penalty is on the inter-task regularization term for reconstructing each task parameter $\boldsymbol{w}_t$.

### 3.1 Asymmetric Transfer from Task to Bases

One critical drawback of (3) is on the severe negative transfer from unreliable models to reliable ones since all task models equally contribute to the construction of latent bases. On the other hand, (4) is not scalable to large number of tasks, and does not learn explicit features. In this section, we provide a novel framework for *asymmetric* multi-task *feature* learning that overcomes the limitations of these two previous approaches, and find an effective way to achieve asymmetric knowledge transfer in deep neural networks while preventing negative transfers.

We start our discussion with the observation of how negative transfer occurs in a common latent bases multi-task learning models as in (3). Suppose that we train a multi-task learning model for three tasks, where the model parameters of each task is generated from the bases $\{\boldsymbol{l}_1, \boldsymbol{l}_2, \boldsymbol{l}_3\}$. Specifically, $\boldsymbol{w}_1$ is generated from $\{\boldsymbol{l}_1, \boldsymbol{l}_3\}$, $\boldsymbol{w}_2$ from $\{\boldsymbol{l}_1, \boldsymbol{l}_2\}$, and $\boldsymbol{w}_3$ from $\{\boldsymbol{l}_2, \boldsymbol{l}_3\}$. Further, we assume that the predictor for task 3 is unreliable and noisy, while the predictors for task 1 and 2 are reliable, as illustrated in Figure 2(a). In such a case, when we train the task predictors in a multi-task learning

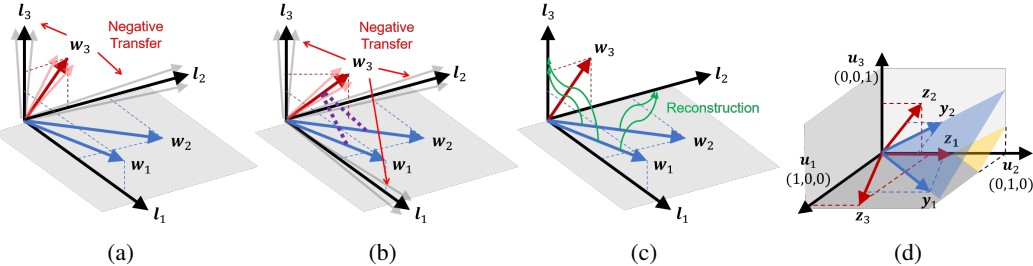

Figure 2: (a) An illustration of negative transfer in common latent bases model. (b) The effects of inter-task $\ell_2$ regularization on top of common latent bases model. (c) Asymmetric task-to-basis transfer. (d) An illustration of ReLU transformation with a bias term.

framework, $\boldsymbol{w}_3$ will transfer noise to the shared bases $\{\boldsymbol{l}_2, \boldsymbol{l}_3\}$, which will in turn negatively affect the models parameterized by $\boldsymbol{w}_1$ and $\boldsymbol{w}_2$.

One might consider the naive combination of the shared basis model (3) and AMTL (4) to prevent negative transfer among latent features where the task parameter matrix is decomposed into $\boldsymbol{LS}$ in (4):

$$\min_{\boldsymbol{L},\boldsymbol{S},\boldsymbol{B}} \sum_{t=1}^{T} \left\{ (1 + \alpha \|\boldsymbol{b}_t^o\|_1) \mathcal{L}(\boldsymbol{Ls}_t; \boldsymbol{X}_t, \boldsymbol{y}_t) + \mu \|\boldsymbol{s}_t\|_1 \right\} + \gamma \|\boldsymbol{LS} - \boldsymbol{LSB}\|_F^2 + \lambda \|\boldsymbol{L}\|_F^2 \quad (5)$$

where $B_{tt} = 0$ for $t = 1, .., T$. However, this simple combination cannot resolve the issue mainly due to two limitations. First, the inter-task transfer matrix $\boldsymbol{B}$ still grows quadratically with respect to $T$ as in AMTL, which is not scalable for large $T$. Second and more importantly, this approach would induce additional negative transfer. In the previous example in Figure 2(b), the unreliable model $\boldsymbol{w}_3$ is enforced to be a linear combination of other reliable models via the matrix $\boldsymbol{B}$ (the purple dashed lines in the figure). In other words, $\boldsymbol{w}_3$ can now affect the clean basis $\boldsymbol{l}_1$ that is only trained by the reliable models in Figure 2(a). As a result, the noise will be transferred to $\boldsymbol{l}_1$, and consequently, to the reliable models based on it. As shown in this simple example, the introduction of inter-task asymmetric transfer in the shared basis MTL (3) leads to more severe negative transfer, which is in contrast to the original intention.

To resolve this issue, we propose a completely new type of regularization in order to prevent the negative transfer from the task predictors to the shared latent features, which we refer to as *asymmetric task-to-basis* transfer. Specifically, we encourage the latent features to be reconstructed by the task predictors' parameters in an asymmetric manner, where we enforce the reconstruction to be done by the parameters of reliable predictors only, as shown in Figure 2(c). Since the parameters for the task predictors are reconstructed from the bases, this regularization can be considered as an autoencoder framework. The difference here is that the consideration of predictor loss result in learning denoising of the representations. We describe the details of our asymmetric framework of task-to-basis transferring in the following subsection.

### 3.2 FEATURE RECONSTRUCTION WITH NONLINEARITY

There are two main desiderata in our construction of asymmetric feature learning framework. First, the reconstruction should be achieved in a non-linear manner. Suppose that we perform linear reconstruction of the bases as shown in Figure 2(c). In this case, the linear span of $\{\boldsymbol{w}_1, \boldsymbol{w}_2\}$ does not cover any of $\{\boldsymbol{l}_1, \boldsymbol{l}_2, \boldsymbol{l}_3\}$. Thus we need a nonlinearity to solve the problem. Second, the reconstruction needs to be done in the feature space, not on the bases of the parameters, in order to directly apply it to a deep learning framework.

We first define notations before introducing our framework. Let $\boldsymbol{X}$ be the row-wise concatenation of $\{\boldsymbol{X}_1, .., \boldsymbol{X}_T\}$. We assume a neural network with a single hidden layer, where $\boldsymbol{L}$ and $\boldsymbol{S}$ are the parameters for the first and the second layer respectively. As for the nonlinearity in the hidden layer, we use rectified linear unit (ReLU), denoted as $\sigma(\cdot)$. The nonnegative feature matrix is denoted as $\boldsymbol{Z} = \sigma(\boldsymbol{XL})$, or $\boldsymbol{z}_i = \sigma(\boldsymbol{Xl}_i)$ for each column $i = 1, .., k$. The task-to-feature transfer matrix is $\boldsymbol{A} \in \mathbb{R}^{T \times k}$. Using the above notations, our asymmetric multi-task feature learning framework is

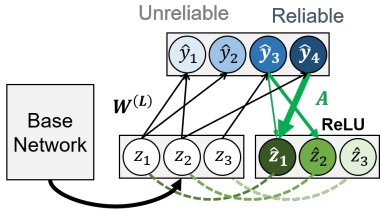

Figure 3: **Deep-AMTFL.** The green lines denote feedback connections with $\ell_{2,1}$ constraints on the features. Different color scales denote different amount of reliabilities (blue) and knowledge transfers from task predictions to features (green).

defined as follows:

$$\min_{\boldsymbol{L},\boldsymbol{S},\boldsymbol{A}} \sum_{t=1}^{T} \left\{ \left(1 + \alpha \|\boldsymbol{a}_t^o\|_1\right) \mathcal{L}(\boldsymbol{L},\boldsymbol{s}_t;\boldsymbol{X}_t,\boldsymbol{y}_t) + \mu \|\boldsymbol{s}_t\|_1 \right\} + \gamma \|\boldsymbol{Z} - \sigma(f(\boldsymbol{Z}\boldsymbol{S})\boldsymbol{A})\|_F^2 + \lambda \|\boldsymbol{L}\|_F^2 . \tag{6}$$

The goal of the autoencoder term is to reconstruct features with model outputs $\boldsymbol{Z}\boldsymbol{S}$, with nonlinear transformation $\sigma(\,\cdot\,;\boldsymbol{A})$. $f$ in the last layer is the nonlinearity for the prediction model (e.g. softmax or logistic function).

We also use ReLU nonlinearity for the reconstruction term as in the original network, since this will allow the reconstruction $\widehat{\boldsymbol{Z}}$ to explore the same manifold of $\boldsymbol{Z}$, thus making it easier to find an accurate reconstruction. In Fig.2(d), for example, the linear span of task output vectors $\{\boldsymbol{y}_1, \boldsymbol{y}_2\}$ forms the blue hyperplane. Transforming this hyperplane with ReLU and a bias term will result in the manifold colored as gray and yellow, which includes the original feature vectors $\{\boldsymbol{z}_1, \boldsymbol{z}_2, \boldsymbol{z}_3\}$.

Since our framework considers the asymmetric transfer in the feature space, we can seamlessly generalize (6) to deep networks with multiple layers. Specifically, the auto-encoding regularization term is formulated at the penultimate layer to achieve the asymmetric transfer. We name this approach Deep-AMTFL:

$$\min_{\boldsymbol{A},\{\boldsymbol{W}^{(l)}\}_{l=1}^{L}} \sum_{t=1}^{T} \left\{ \left(1 + \alpha \|\boldsymbol{a}_t^o\|_1\right) \mathcal{L}_t + \mu \left\|\boldsymbol{w}_t^{(L)}\right\|_1 \right\}$$
$$+ \gamma \left\|\sigma\big(f(\boldsymbol{Z}\boldsymbol{W}^{(L)})\boldsymbol{A}\big) - \boldsymbol{Z}\right\|_F^2 + \lambda \sum_{l=1}^{L-1} \left\|\boldsymbol{W}^{(l)}\right\|_F^2 , \tag{7}$$

where $\boldsymbol{W}^{(l)}$ is the weight matrix for the $l^{th}$ layer, with $\boldsymbol{w}_t^{(L)}$ denoting the $t^{th}$ column vector of $\boldsymbol{W}^{(L)}$, and

$$\boldsymbol{Z} = \sigma(\boldsymbol{W}^{(L-1)}\sigma(\boldsymbol{W}^{(L-2)}\ldots\sigma(\boldsymbol{X}\boldsymbol{W}^{(1)})))$$
$$\mathcal{L}_t := \mathcal{L}(\boldsymbol{w}_t^{(L)}, \boldsymbol{W}^{(L-1)}, .., \boldsymbol{W}^{(1)}; \boldsymbol{X}_t, \boldsymbol{y}_t),$$

are the hidden representations at layer $L-1$ and the loss for each task $t$. See Figure (3) for the description.

**Loss functions** The loss function $\mathcal{L}(\boldsymbol{w};\boldsymbol{X},\boldsymbol{y})$ could be any generic loss function. Throughout the paper, we mainly consider the two most popular instances. For regression tasks, we use the squared loss: $\mathcal{L}(\boldsymbol{w}_t;\boldsymbol{X}_t,\boldsymbol{y}_t) = \frac{1}{N_t}\|\boldsymbol{y}_t - \boldsymbol{X}_t\boldsymbol{w}_t\|_2^2 + \delta/\sqrt{N_t}$. For classification tasks, we use the logistic loss: $\mathcal{L}(\boldsymbol{w}_t;\boldsymbol{X}_t,\boldsymbol{y}_t) = \frac{1}{N_t}\sum_{i=1}^{N_t}\{y_{ti}\log\sigma(\boldsymbol{x}_{ti}\boldsymbol{w}_t) + (1-y_{ti})\log(1-\sigma(\boldsymbol{x}_{ti}\boldsymbol{w}_t))\} + \delta/\sqrt{N_t}$, where $\sigma$ is the sigmoid function. Note that we augment the loss terms with $\delta/\sqrt{N_t}$ to express the imbalance for the training instances for each task, where $\delta$ is a parameter to tune.

# 4 EXPERIMENTS

## 4.1 SHALLOW MODELS - FEEDFORWARD NETWORKS

We first validate our shallow AMTFL model (6) on synthetic and real datasets with shallow neural networks. Note that shallow models use one-vs-all logistic loss for multi-class classification.

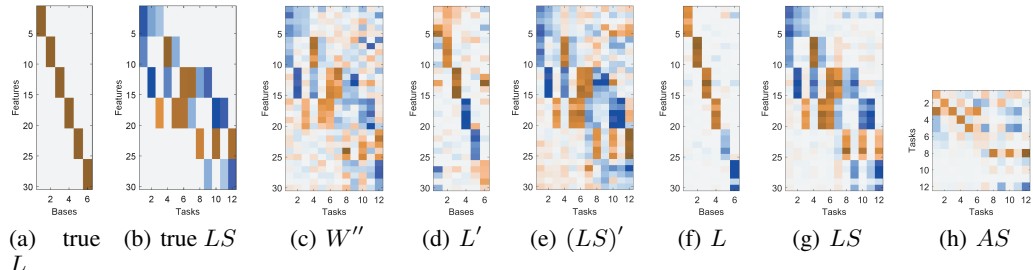

| (a) true $L$ | (b) true $LS$ | (c) $W''$ | (d) $L'$ | (e) $(LS)'$ | (f) $L$ | (g) $LS$ | (h) $AS$ |

Figure 4: **Visualization of the learned features and paramters on the synthetic dataset.** (a-b) True parameter for generating the dataset. (c) Reconstructed parameters from AMTL (d-e) Reconstructed parameters from Go-MTL. (f-h) Reconstructed parameters from AMTFL.

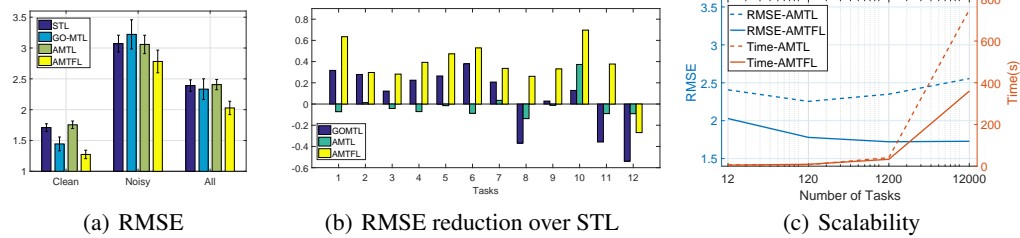

| (a) RMSE | (b) RMSE reduction over STL | (c) Scalability |

Figure 5: **Results of synthetic dataset experiment.** (a) Average RMSE for clean/noisy/all tasks. (a) Per-task RMSE reduction over STL. (b) RMSE and training time for increasing number of tasks.

**Baselines and our models**

**1) STL.** A linear single-task learning model.

**2) GO-MTL.** A feature-sharing MTL model from Kumar & Daume III (2012), where different task predictors share a common set of latent bases (3).

**3) AMTL.** Asymmetric multi-task learning model (Lee et al., 2016), with inter-task knowledge transfer through a parameter-based regularization (4).

**4) NN.** A simple feedforward neural network with a single hidden layer.

**5) MT-NN.** Same as NN, but with each task loss divided by $N_t$, for balancing the task loss. Note that this model applies $\ell_1$-regularization at the last fully connected layer.

**6) AMTFL.** Our asymmetric multi-task feature learning model with feedback connections (6).

### 4.1.1 SYNTHETIC DATASET EXPERIMENT

We first check the validity of AMTFL on a synthetic dataset. We first generate six 30-dimensional true bases in Figure 4(a). Then, we generate parameters for 12 tasks from them with noise $\epsilon \sim \mathcal{N}(0, \sigma)$. We vary $\sigma$ to create two groups based on the noise level: easy and hard. Easy tasks have noise level of $\sigma = 1$ and hard tasks have noise level of $\sigma = 2$. Each predictor for easy task $\boldsymbol{w}_t$ combinatorially picks two out of four bases - $\{\boldsymbol{l}_1, .., \boldsymbol{l}_4\}$ to linearly combine $\boldsymbol{w}_t \in \mathbb{R}^{30}$, while each predictor for hard task selects among $\{\boldsymbol{l}_3, .., \boldsymbol{l}_6\}$. Thus the bases $\{\boldsymbol{l}_3, \boldsymbol{l}_4\}$ overlap both easy and hard tasks, while other bases are used exclusive by each group in Figure 4(b). We generate five random train/val/test splits for each group - $\{50/50/100\}$ for easy tasks and $\{25/25/100\}$ for hard tasks.

For this particular dataset, we implement all base models as neural networks to better compare with AMTFL. We add in $\ell_1$ to $\boldsymbol{L}$ for all models for better reconstruction of $\boldsymbol{L}$. We remove ReLU at the hidden layer in AMTFL since the features are linear[1]. All the hyper-parameters are found with separate validation set. For AMTL, we remove the nonnegative constraint on $\boldsymbol{B}$ due to the characteristic of this dataset.

---

[1] We avoid adding nonlinearity to features to make qualitative analysis much easier.

Table 1: Performance of the baselines and our asymmetric multi-task feature learning models. We report the RMSE for regression and mean classification error(%) for classification, along with the standard error for 95% confidence interval. Performance of AMTL on AWA-A is not available, since multi-label classification is not implemented in provided codes.

| Models | AWA-A | MNIST | School | Room |
|--------|-------|-------|--------|------|
| STL | 37.6±0.5 | 14.8±0.6 | 10.16±0.08 | 45.9±1.4 |
| Go-MTL | 35.6±0.2 | 14.4±1.3 | **9.87±0.06** | 47.1±1.4 |
| AMTL | N/A | 12.9±1.4 | 10.13±0.08 | 40.8±1.5 |
| NN | 26.3±0.3 | 10.1±1.0 | 9.89±0.03 | 46.1±2.6 |
| MT-NN | 26.2±0.3 | 8.84±1.1 | 9.91±0.04 | 41.7±3.0 |
| AMTFL | **25.2±0.3** | **8.52±1.1** | 9.89±0.09 | **40.1±1.9** |

We first check whether AMTFL can accurately reconstruct the true bases in Figure 4(a). We observe that $L$ learned by AMTFL in Figure 4(f) more closely resembles the true bases than $L'$ reconstructed using Go-MTL in Figure 4(d)), which is more noisy. The reconstructed $W = LS$ from AMTFL in Figure 4(g), in turn, is closer to the true parameters than $W' = (LS)'$ generated with Go-MTL in Figure 4(e) and $W''$ from AMTL in Figure 4(c) for both easy and hard tasks. Further analysis of the inter-task transfer matrix $AS$ in Figure 4(h) reveals that this accurate reconstruction is due to the asymmetric inter-task transfer, as it shows no transfer from hard to easy tasks, while we see significant amount of transfers from easy to hard tasks.

Quantitative evaluation result in Figure 5(a) further shows that AMTFL significantly outperforms existing MTL methods. AMTFL obtains lower errors on both easy and hard tasks to STL, while Go-MTL results in even higher errors than those obtained by STL on hard tasks. We attribute this result to the negative transfer from other hard tasks. AMTFL also outperforms AMTL by significant margin, maybe because it is hard for AMTL to find meaningful relation between tasks in case of this particular synthetic dataset, where data for each task is assumed to be generated from the same set of latent bases. Also, a closer look at the per-task error reduction over STL in Figure 5(b) shows that AMTFL effectively prevents negative transfer while GO-MTL suffers from negative transfer, and make larger improvements than AMTL. Further, Figure 5(c) shows that AMTFL is more scalable than AMTL, both in terms of error reduction and training time, especially when we have large number of tasks. One thing to note is that, for AMTFL, the error goes down as the number of tasks increases. This is a reasonable result, since the feature reconstruction using the task-specific model parameters will become increasingly accurate with larger number of tasks.

### 4.1.2 REAL DATASET EXPERIMENT

We further test our model on one binary classification, one regression, and two multi-class classification datasets, which are the ones used for experiments in (Kumar & Daume III, 2012; Lee et al., 2016). We report averaged performance of each model on five random splits for all datasets.

**1) AWA-A:** This is a classification dataset (Lampert et al., 2009) that consists of $30,475$ images, where the task is to predict *85 binary attributes* for each image describing a single animal. The feature dimension is reduced from 4096 (Decaf) to $500$ by using PCA. The number of instances for train, validation, and test set for each task is $1080$, $60$, and $60$, respectively. We set the number of hidden neurons to $1000$ which is tuned on the base NN.

**2) MNIST:** This is a standard dataset for classification (LeCun et al., 1998) that consists of $60,000$ training and $10,000$ test images of $28 \times 28$ that describe 10 handwritten digits (0-9). Following the procedure of Kumar & Daume III (2012), feature dimension is reduced to $64$ by using PCA, and 5 random 100/50/50 splits are used for each train/val/test. We set the number of hidden neurons to $200$.

**3) School:** This is a regression dataset where the task is to predict the exam scores of $15,362$ student's from 139 schools. Prediction of the exam score for each school is considered as a single task. The splits used are from Argyriou et al. (2008) and we use the first 5 splits among 10. We set the number of hidden neurons to 10 or 15.

**4) Room:** This is a subset of the ImageNet dataset (Deng et al., 2009) from Lee et al. (2016), where the task is to classify $14,140$ images of 20 different indoor scene classes. The number of train/val

Table 2: Average per-class error (%) of baseline deep networks and our deep asymmetric multi-task feature learning models.

|  | CIFAR-100 | AWA-C | ImageNet-352 |
| --- | --- | --- | --- |
| CNN | 19.65 | 11.36 | 66.54 |
| MT-CNN | 19.65 | 10.54 | 65.69 |
| Deep-AMTL | 19.51 | 10.27 | 65.61 |
| Deep-AMTFL | **19.20** | **9.96** | **64.49** |

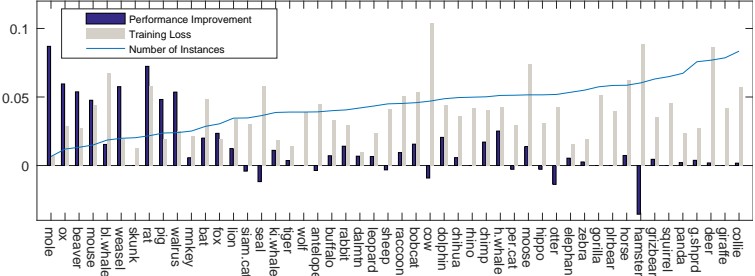

Figure 6: Per-class performance improvements on AWA dataset.

instances varies from 30 to over 1000, while test set has 20 per each class. The feature dimension is reduced from 4096 (Decaf) to 500 by using PCA. We set the number of hidden neurons to 1000.

Table 1 shows the results on the real datasets. As expected, the AMTFL (6) outperforms the baselines on most datasets.

## 4.2 DEEP MODELS - CONVOLUTIONAL NETWORKS

Next, we validate our Deep-AMTFL (7) with deep CNNs for end-to-end image classification. Note that deep models use softmax loss instead of one-vs-all logistic loss.

**Baselines and our models**

**1) CNN:** Base convolutional neural network.

**2) MT-CNN:** CNN with loss for each class $t$ divided by $N_t$, the number of positive instance for each class.

**3) Deep-AMTL:** The same as Multitask CNN, but with the asymmetric multi-task learning objective in Lee et al. (2016) replacing the original loss. Note that the model is deep version of (5).

**4) Deep-AMTFL:** Our deep asymmetric multi-task feature learning model with feedback connections (7).

**Datasets and base networks**

**1) CIFAR-100:** This dataset (Krizhevsky & Hinton, 2009) consists of $60,000$ images from $100$ object categories. The predefined train/test split is $500/100$ per class. We use Wide Residual Network (Zagoruyko & Komodakis, 2016) for the base network, with 28 layers and the widening factor of $10$ [2].

**2) AWA-C:** This is the same AWA dataset used in the shallow model experiments, but we use $50$ animals *class* labels instead of binary attributes. We use all the raw pixel images, with the number of class instances being quite imbalanced. We fined-tune ResNet (He et al., 2016) with $50$ layers trained on ImageNet-1K (Deng et al., 2009) dataset for the base network.

**3) ImageNet-352:** This is a subset of the ImageNet 22K dataset (Deng et al., 2009), which contains $352$ classes selected from the entire $22K$ classes. We deliberately created the dataset to be largely

---

[2] We extensively tuned the WRN to reproduce the results in (Zagoruyko & Komodakis, 2016), but could not reproduce it; this incident have been reported by various researchers.

imbalanced over classes, ranging from 2 to $1,044$. The base network is the same as in AWA-C dataset.

**Experimental Setup**   For the implementation of all the baselines and our deep models, we use Caffe (Jia, 2013) framework. We plan to make our codes public for reproduction, if the paper gets accepted. For CNN and MT-CNN, we train the model from scratch, while rest of the models are finetuned from MT-CNN for expedited training. More detailed experimental setup is in the supplementary file.

**Quantitative evaluation**   We report the average per-class classification performances of baselines and our models in Table 2. Our Deep-AMTFL outperforms all baselines, including Deep-AMTL, which shows the effectiveness of asymmetric autoencoder in deep learning framework.

To see where the performance improvement comes from, we further report the per-task (class) performance improvement of our AMTFL over the base CNN on AWA dataset in Figure 6, along with class-specific loss and number of instances per class. AMTFL outperforms the base CNN on $41$ classes out of all $50$ classes, and does not degenerate performances except for few classes. This result suggests that the performance improvement mostly comes from the suppression of negative transfer. The classes with large performance degeneration, *cow* and *hamster*, are the ones with large training loss; this is a somewhat expected result since the proxy term $\delta/\sqrt{N_t}$, which may be unreliable at times as the measure of true risk.

### 4.3   APPLICATION TO TRANSFER LEARNING

For this experiment, we use the AWA dataset, which is a standard dataset for transfer learning that provides source/target task class split. The source dataset contains $40$ animal classes including *grizzly bear*, *hamster*, *blue whale*, and *tiger*, and the target dataset contains $10$ animal classes, including *giant panda*, *rat*, *humpback whale*, and *leopard*. Thus the tasks in two datasets exhibit large degree of relatedness. We train baseline networks and our Deep-AMTFL model on the source dataset, and trained a softmax classifier on the layer just before the softmax layer of the original network while maintaining all other layers to be fixed, for the classification of the target dataset.

Table 3: Classification error(%) of the baselines and our model on the transfer learning task. Source networks denote types of networks that is trained on the source dataset with 40 classes, and Target accuracy is the accuracy of the softmax classifier on 10 target classes trained on the representations obtained at the layer just below the softmax layer of the source network.

| Source Network | Target Accuracy |
|----------------|-----------------|
| CNN | 5.00 |
| Deep-AMTL | 5.00 |
| Deep-AMTFL | **4.33** |

## 5   CONCLUSION

We propose a novel deep asymmetric multi-task feature learning framework that can effectively prevent negative transfer resulting from symmetric influences of each task in feature learning. By introducing an asymmetric feedback connections in the form of autoencoder, our AMTFL enforces the participating task predictors to asymmetrically affect the learning of shared representations based on task loss. We perform extensive experimental evaluation of our model on various types of tasks on multiple public datasets. The experimental results show that our model significantly outperforms both the symmetric multi-task feature learning and asymmetric multi-task learning based on inter-task knowledge transfer, for both shallow and deep frameworks.

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
