# OpenReview forum: "Deep Asymmetric Multi-task Feature Learning"
_ICLR.cc/2018/Conference — Reject_

### Official Review · AnonReviewer3 · 2017-11-26
**a borderline paper**

**Rating:** 6
**Confidence:** 4

**Review:**

This paper presents a deep asymmetric multi-task feature learning method (Deep-AMTFL).

One concern is that the high similarity between the proposed Deep-AMTFL and an existing AMTL method. Even though AMTL operates on task relations and Deep-AMTFL is on feature learning, the main ideas of both methods are very similar, that is, tasks with higher training losses will contribute less to other tasks' model or feature representations. Even though the regularizers seem a bit different, the large similarity with AMTL decreases the novelty of this work.

In real-world experiments, it is better to show the difference of learned features among the proposed Deep-AMTFL and other baselines.

A minor problem: the last sentence in page 3 is incomplete.

---

### Official Review · AnonReviewer2 · 2017-11-27

**Rating:** 3
**Confidence:** 4

**Review:**

Summary: The paper proposes a multi-task feature learning framework with a focus on avoiding negative transfer. The objective has two kinds of terms to minimise: (1) The reweighed per-task loss, and (2) Regularisation. The new contribution is an asymmetric reconstruction error in the regularisation term, and one parameter matrix in the regulariser influences the reweighing of the pre-task loss.

Strength:
The method has some contribution in dealing with negative transfer. The experimental results are positive.
Weakness:
Several issues in terms of concept, methodology, experiments and analysis.

Details:
1. Overall conceptual issues.
1.1. Unclear motivation re prior work. The proposed approach is motivated by the claim that GO-MTL style models assumes symmetric transfer where bad tasks can hurt good tasks. This assertion seems flawed. The point of grouping/overlap in “GO”-MTL is that a “noisy”, “hard”, or “unrelated" task can just take its own latent predictor that is disjoint from the pool of predictors shared by the good/related tasks.
Correspondingly, Fig 2 seems over-contrived. A good GO-MTL solution would assign the noisy task $w_3$ its own latent basis, and let the two good tasks share the other two latent bases.

1.2  Very unclear intuition of the algorithm. In the AMTFL, task asymmetry is driven by the per-task loss. The paper claims this is because transfer must go from easy=>hard to avoid negative transfer. But this logic relies on several questionable assumptions surrounding conflating the distinct issues of difficulty and relatedness: (i) There could be several easy tasks that are totally un-related. One could construct synthetic examples with data that are trivially separable (easy) but require unrelated or orthogonal classifiers. (ii) A task could appear to be “easy" just by severe overfitting, and therefore still be detrimental to transfer despite low loss. (iii) A task could be very "difficult" in the sense of high loss, but it could still be perfectly learned in the sense of finding the ideal "ground-truth” classifier, but for a dataset that is highly non-separable in the provided feature-space. Such a perfectly learned classifier may still be useful to transfer despite high loss. (iv) Analogous to point (i), there could be several “difficult” tasks that are indeed related and should share knowledge. (Since difficult/high loss != badly learned as mentioned before). Overall there are lots of holes in the intuitive justification of the algorithm.

2. Somewhat incremental method.
3.1 It’s a combination of AMTL (Lee 2016) and vanilla auto encoder.

3. Methodology issues:
3.1 Most of the explanation (Sec 3-3.1) is given re: Matrix B in Eq.(4) (AMTL method’s objective function). However the final proposed model uses matrix A in Eq.(6) for the same purpose of measuring the amount of outgoing transfers from task $t$ to all other tasks. However in the reconstruction loss, they work in very different ways: matrix B is for the reconstruction of model parameters, while matrix A is for the reconstruction of latent features. This is a big change of paradigm without adequate explanation. Why is it still a valid approach?
3.2 Matrix B in the original paper of AMTL (Eq.(1) of Lee et al., 2016) has a constraint $B \geq 0$, should matrix A have the same constraint? If not, why?
3.3 Question Re: the |W-WB| type assumption for task relatedness. A bad task could learn an all-zero vector of outgoing related ness $b^0_t$ so it doesn’t directly influence other tasks in feed-forward sense. But hat about during training? Does training one task’s weights endup influencing other tasks’s weights via backprop? If a bad task is defined in terms of incoming relatedness from good tasks, then tuning the bad task with backprop will eventually also update the good tasks? (presumably detrimentally).

4. Experimental Results not very strong.
4.1 Tab 1: Neural Network NN and MT-NN beat the conventional shallow MTL approaches decisively for AWA and MNIST.  The difference between MT-NN and AMTFL is not significant. The performance boost is more likely due to using NNs rather than the proposed MTL module. For School, there is not significant difference between the methods. For ImageNet-Room AMTL and AMTFL have overlapping errors. Also, a variant of AMTL (AMTL-imbalance) was reported in Lee’2016, but not here where the number is $40\pm1.71$.
4.2 Tab 2: The “real” experiments are missing state of the art competitors. Besides a deep GO-MTL alternative, which should be a minimum,  there are lots of deep MTL state of the art: Misra CVPR’16 , Yang ICLR’17, Long arXiv/NIPS’17 Multilinear Relationship Nets,  Ruder arXiv’17 Sluice Nets, etc.

5. Analysis
5.1 The proposed method revolves around the notion of “noisy”/“unrelated”/“difficult” tasks. Although the paper conflates them, it may still be a useful algorithm in practice. But it in this case it should devise much better analysis to provide insight and convince us that this is not a fatal oversimplification: What is the discovered relatedness matrix in some benchmarks? Does the discovered relatedness reflect expert knowledge where this is available? Is there a statistically significant correlation between relatedness and task difficulty in practice? Or between relatedness and degree of benefit from transfer, etc? But this is hard to do cleanly as even if the results show a correlation between difficulty and relatedness, it may just be because that’s how relatedness is defined in the proposed algorithm.

---

### Official Review · AnonReviewer1 · 2017-11-27
**This paper addresses a multi-task feature learning setting, but the objective lacks clarity and the experiments are not convincing.**

**Rating:** 5
**Confidence:** 4

**Review:**

This paper addresses multi-task feature learning, i.e. learning representations that are common across multiple related supervised learning tasks. The paper is not clearly written, so I outline my interpretation on what is the main idea of the manuscript.

The authors rely on two prior works in multi-task learning  that explore parameter sharing (Lee et al, 2016) and subspace learning (Kumar & Daume III 2012) for multi-task learning.
1) The work of Lee et al 2016 is based on the idea of transferring information through weight vectors, where each task parameter can be represented as a sparse combination of other related task parameters. The interpretation is that negative transfer is avoided because only subset of relevant tasks is considered for transfer. The drawback is the scalability of this approach.
2) The second prior work is Kumar & Daume III 2012 (and also an early work of Argyrio et al 2008) that is based on learning a common feature representation. Specifically, the main assumption is that tasks parameters lie in a low-dimensional subspace, and parameters of related tasks can be represented as linear combinations of a small number of common/shared latent basis vectors in such subspace.  Subspace learning could help to scale up to many tasks.

The authors try to combine together the ideas/principles in these previous works and propose a sparse auto encoder model for multi-task feature learning with (6) (and (7)) as the main learning objectives for training an autoencoder.

- I couldn’t fully understand the objective in (6) and how exactly it is related to the previous works, i.e. how the relatedness and easyness/hardness of tasks is measured; where does f enter in the autoencoder network structure?
- The empirical evaluations are not convincing. In the real experiments with image data, only decaf features were used as input to the autoencoder model. Why not using raw input image? Moreover all input features where projected to a lower dimensional space using PCA before inputing to the autoencoder. Why? In fact, linear PCA can be viewed as an autoencoder model with linear encoder and decoder (so that the squared error reconstruction loss between a given sample and the sample reconstructed by the autoencoder is minimal (Bishop, 2006)). Then doing PCA before training an autoencoder is not motivated.

-Writing can be improved. The introduction primarily criticizes  the approach of Lee et al, 2016 called Assymetric Multi-task Learning. It would be nicer if the introduction sets the background and covers different approaches/aspects/conditions of negative transfer in transfer learning/multi-task learning setting. The main learning objective (6) should be better explained.

-Conceptual picture is a bit lacking. Striped hyena is used as an example of unreliable noisy data (source of negative transfer) when learning the attribute classifier "stripes". One might argue that visually, striped hyena is as informative as white tigers. Perhaps one could use a different (less striped) animal, e.g. raccoon.

---

### Decision · Program_Chairs · 2018-01-29
**ICLR 2018 Conference Acceptance Decision**

**Decision:**

Reject

**Comment:**

The paper proposes a multitask deep learning method (called Deep-AMFTL) for preventing negative transfer. Despite some positive experimental results, the contribution of the paper is not sufficient for publication at ICLR due to several issues: similarity between the proposed method and existing method (e.g., AMTL), unclear rationale/intuition of the proposed model, clarity of presentation, technical formulation, and limited empirical evaluations (see reviewer comments for details). No author rebuttal was submitted.